# Optimizing Autonomous Vehicle Communication through an Adaptive Vehicle-to-Everything (AV2X) Model: A Distributed Deep Learning Approach

Radwa Ahmed Osman 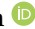

Basic and Applied Science Department, College of Engineering and Technology, Arab Academy for Science and Technology (AAST), Alexandria 1029, Egypt; adwa.ahmed@aast.edu

**Abstract:** Autonomous intelligent transportation systems consistently require effective and secure communication through vehicular networks, enabling autonomous vehicle communication. The reduction of traffic congestion, the alerting of approaching emergency vehicles, and assistance in low visibility traffic are all made possible by effective communication between autonomous vehicles and everything (AV2X). Therefore, a new adaptive AV2X model is proposed in this paper to improve the connectivity of vehicular networks. This proposed model is based on the optimization method and a distributed deep learning model. The presented approach optimizes the inter-vehicle location if required for ensuring effective communication between the autonomous vehicle (AV) and everything (X) using the Lagrange optimization algorithm. Furthermore, the system is evaluated in terms of energy efficiency and achievable data rate based on the optimal inter-vehicle position to show the significance of the proposed approach. To meet the stated goals, the ideal inter-vehicle position is predicted using a distributed deep learning model by learning from mathematically generated data and defined as a restricted optimization problem using the Lagrange optimization technique to improve communication between AV2X under various environmental conditions. To demonstrate the efficiency of the suggested model, the following characteristics are considered: vehicle dispersion, vehicle density, vehicle mobility, and speed. The simulation results show the significance of the proposed model in terms of energy efficiency and achievable data rate compared with other proposed models.

**Keywords:** autonomous vehicles; vehicular communication; vehicle-to-everything (V2X); cooperative communication; energy efficiency; achievable data rate

## 1. Introduction

Current intelligent transportation systems remain in the development stage due to the complexity of their numerous components, including the control of urban roads, pedestrians, and vehicles [1]. On the other hand, intelligent driving support systems, which are suggested based on the active safety principle and attempts to shorten drivers' reaction times, have been evolving quickly. Autonomous vehicles (AVs), which are part of intelligent transportation, are starting to appear on commercial roads and are moving towards full autonomy [2]. In order to drive safely and effectively today, AVs rely on sensors like cameras and LiDAR to monitor the road [3]. However, with the presence of other AVs on the road, they can enhance their capabilities through vehicle-to-vehicle (V2V) communication. By sharing real-time information among AVs, driving becomes safer and more efficient [4]. Through this V2V communication, AVs can exchange information, predict one another's movements, and react appropriately, improving traffic flow and boosting road safety. The incorporation of V2V communication is becoming increasingly important as AV technology develops if completely autonomous and intelligent transportation systems are to be realized [5].

As AVs gain popularity and move closer to universal use, the fundamental necessity for smooth communication and connectivity becomes more obvious. Vehicle-to-Infrastructure (V2I) and Vehicle-to-Vehicle (V2V) communications play a role here by improving typical autonomous driving duties [6]. Additionally, they will also connect AVs to the Internet of Things (IoT) as a whole, enabling for a more connected environment [7]. IoT is a term without a clear definition; however, it refers to the networking of everyday objects, typically through wireless links [8]. It is worth mentioning the 5G technology, which promotes autonomous driving and improves vehicle connectivity. Different techniques have been developed to enhance the performance of 5G networks [9,10]. Furthermore, Beyond 5G (B5G) and 6G communication technologies are widely regarded as the future of V2V communications because of their high throughput, high bandwidth, and low latency [11]. Machine learning (ML) techniques, particularly deep neural networks, are currently the foundation of advanced technology autonomous driving systems [12]. Deep neural networks are utilized in autonomous cars for many different purposes, such as localization, path planning, and perception [13]. However, in order to operate effectively, these algorithms need sources of informative data, which vehicular communications can offer.

Furthermore, the relevance of employing optimization algorithms in many types of applications to address a number of concerns and problems should be emphasized [14,15]. Optimization algorithms are critical in the development and operation of autonomous vehicles, allowing them to operate more efficiently, safely, and reliably in a variety of real-world circumstances. Future progress and broad adoption of autonomous vehicles will be further aided by ongoing improvements in optimization techniques [16]. In the previous work proposed in [4], the authors focused on enhancing the connectivity between AVs and everything based on finding the best relay position using an optimization technique. Consequently, this proposed model has been enhanced to a new effective model by using a deep learning model with the optimization technique. As a result, the following are the main contributions of this article:

- The suggested approach mixes an optimization technique with a deep learning algorithm to create a new, effective way to improve connectivity between AVs and everything (X).
- To determine the reliability and efficiency of the communication between AVs and everything, an optimization problem is established utilizing the Lagrange optimization technique and a 1D Convolutional Neural Network (1D-CNN) network.
- The suggested approach attempts to improve communication between AVs and destinations (AV2X) in terms of energy efficiency (*EE*) and achievable data rate (*R*). This can be accomplished by identifying the shortest possible distance between AVs and anything in order to maximize total EE and R. Alternately, other factors should be taken into account like transmission power, interference distances that could arise from existing transmission devices sharing the same spectrum, the necessary signal-to-interference-plus-noise ratio ($SINR_{th}$), and path loss.
- Through the deep learning model, AVs will be able predict the maximum suitable permissible transmission distance while taking into consideration various environmental conditions.
- The proposed approach is examined in terms of energy efficiency and overall achievable data rate under various environmental conditions, such as transmission power, transmission ranges, and needed $SINR_{th}$ values. These discoveries allow for the enhancement of AV networks' performance.

The following is the order of the sections: The relevant work will be presented in Section 2. Section 3 will go over the specific proposed strategy in depth. The experimental and analytical work of the suggested approach will be displayed in Section 4. Finally, Section 5 will present the conclusion.

## 2. Related Work

Fifth-generation wireless communications (5G) and artificial intelligence (AI) are thought to be two separate research fields. Artificial intelligence and next-generation wireless communication technologies have been specifically developed to meet applications at this convergence of the creation of an AI-based Vehicle-to-Everything (AI-V2X) system for intelligent cities and autonomous vehicles that combine artificial intelligence with 5G networking technology. To improve traffic performance and safety, ref. [17] proposed a new emergency vehicle route-clarifying technique based on vehicle-to-vehicle (V2V) communication. The technique is intended to locate the nearest car with which to contact and clear the road lane, ensuring that the emergency vehicle arrives at its location with as little travel time as possible. To improve Quality-of-Service (QoS) in high-mobility vehicular networks, ref. [18] developed a cooperative communication strategy that combines V2I and V2V approaches. It increased system reliability and efficiency in a variety of environmental circumstances by optimizing key QoS performances through adaptive transmission scheme selection. Ref. [19] looked at ways to improve vehicle-to-vehicle communication (V2V) using 5G technology, which provides faster data rates and more bandwidth than the DSRC (5.9 GHz) system it replaces. The study looked at how sand and dust affected the communication paths lost for both DSRC and 5G frequency bands in urban and highway settings, and it proposed a new link margin model to calculate their effects. Moreover, ref. [20] developed a deep learning approach to improve Vehicle-to-Everything (V2X) communication in vehicular networks, with the goal of determining the optimal interference power for increased connectivity, QoS, and communication link reliability. The findings confirmed the model's ability to improve road traffic information efficiency and safety through increased V2X communication.

In the context of connected and autonomous vehicles, this convergence of technologies attempts to maximize the potential of both AI and 5G to improve communication and performance. The proposed strategy in [21] aimed to improve driving performance, comfort, and safety by utilizing data-driven approaches and real-time communication, leading to appreciable advancements in transportation, traffic monitoring, V2X communication development, driving comfort, and a decrease in traffic congestion. Furthermore, ref. [22] suggested a 5G Vehicle-to-Everything (V2X) Intelligent Intrusion Detection System (IIDS) for connected and autonomous vehicles (CAVs) based on a modified Convolutional Neural Network (CNN). The IIDS detected attacks with a remarkable 98% accuracy, effectively increasing security in the Internet of Vehicles (IoV) system by identifying and classifying rogue AVs. Moreover, ref. [23] provided a comprehensive overview of 5G New Radio (NR) in cellular vehicle-to-everything (V2X) systems for future connected autonomous communication networks. The article covered various aspects, including the physical layer, sidelink communication, architecture flexibility, security, privacy mechanisms, precise positioning techniques, and the potential of machine learning for performance enhancement. Additionally, it highlighted 5G NR's configuration to support advanced V2X use cases. Additionally, in [24] a deep learning-based traffic safety solution for a mixed-autonomous/manual Intelligent Transportation System (ITS) was proposed, leveraging 5G technology. By utilizing driving trajectory and natural-driving datasets, the approach achieved higher real-time intention identification rates for lane changes, resulting in improved accuracy and safety in mixed traffic scenarios.

It is worth mentioning that other techniques were developed to improve the performance of AVs in different manners. Ref. [25] presented a secure user authentication mechanism for self-driving cars operating in a 5G network. Using non-interactive zero-knowledge proof technology and a physical uncloning function, the protocol provides mutual authentication and key agreement without divulging sensitive information. A formal security study validates its ability to withstand hostile attacks while remaining efficient. The study presented in [26] depicted real-world vehicle communication scenarios that make use of 5G technology. Vehicles exchanged sensor and controlled data with one another, their surroundings, and their Digital Twins, offering insights on the practicality

of 5G Non-Standalone Architecture for certain communication scenarios, including existing latency and throughput limitations in real-world conditions. Furthermore, ref. [27] examined how 5G+ technology could be applied to the transportation sector, with a particular emphasis on driverless cars and intelligent transportation systems. By resolving issues and using 5G+ capabilities, SDN, and intelligent infrastructures to suit future user expectations for security, low latency, and energy efficiency, the adaptable framework that was suggested sought to improve smart mobility. The model proposed in [28] addressed spectrum and power allocation in V2X communication with unknown channels while taking into account various elements such as Doppler shift and inaccurate CSI. Bernstein Approximation-based and self-learning are two strong resource allocation methodologies that are robust approaches for maximizing cellular user equipment (CUE) capacity while assuring SINR for CUEs and reliability for vehicular user equipment (VUEs).

Despite existing approaches in the literature, there is still a need for further investigation to improve the performance of autonomous vehicles (AVs) through the implementation of Lagrange optimization algorithms and deep learning techniques. This study aims to identify the essential parameters and conditions that can enhance AVs' energy efficiency and achievable data rate, ensuring a more efficient and reliable system. An analytical optimization technique, combined with a distributed deep learning model for AV networks, is proposed to enable AVs to adapt the effective required transmission distance, leading to enhanced efficiency and reliability in AV networks. The performance of the AV system is evaluated based on energy efficiency and achievable data rate. Table 1 compares the uniqueness of the proposed model to previous comparable research articles. Therefore, the proposed model is focused on enhancing energy efficiency and achievable data rate due to the specific requirements and limitations of Vehicle-to-Vehicle (V2V) communication. V2V communication entails vehicles to communicate with one another to improve road safety, traffic efficiency, and other applications. However, because V2V devices are frequently battery-powered, minimizing energy usage is critical for extending the operating lifetime of these devices without frequent battery changes or recharges. The correct combination of data rate and energy efficiency in V2V communication means that safety-essential information may be exchanged quickly while preserving energy resources, making it a critical concern in this specific application.

**Table 1.** Comparison between different related works and the proposed model.

| | Technique Used | Optimization Problem | Deep Learning Technique | Metric for System Evaluation | Investigation Scenario |
|---|---|---|---|---|---|
| [4] | Multihop relaying between AV2X; direct AV2X and V2V communications. | Improving system QoS by determining the best autonomous inter-vehicle position from which to connect with or relay information to any destination. | N/A | • Best relay-vehicle position between AV2X.<br>• Packet delivery ratio.<br>• Throughput.<br>• Packet loss rate.<br>• Average delivery latency. | Communication between AV2X is relayed with direct communication between V2V. |
| [17] | Multihop V2V communication. | Finding the maximum distance between the emergency vehicle and the nearest vehicle through Lagrance optimization. | N/A | • Packet delivery ratio.<br>• Average end-to-end-delay | V2V communication is used to find the nearest vehicle to communicate with in order to get information about road traffic conditions. |
| [18] | Cooperative communications for a combined V2I with V2V approach. | Minimizing total energy consumed per bit given a target outage probability, or maximizing end-to-end throughput | N/A | • Energy consumption<br>• Throughput<br>• Packet delivery ratio<br>• Packet loss rate<br>• Average end-to-end-delay | • Direct V2I communication<br>• Multi-hop V2I communication<br>• Cooperative V2I communication |

**Table 1.** *Cont.*

| | Technique Used | Optimization Problem | Deep Learning Technique | Metric for System Evaluation | Investigation Scenario |
|---|---|---|---|---|---|
| [19] | Direct V2V communication. | N/A | N/A | • Received Packets<br>• Collision distance | Investigate the effect of dust and sand on the quality of the received signal throughout the communication between V2V. |
| [20] | A distributed deep learning technique is used to control interference power. | Maximizing V2X system performance; determining the optimal needed interference power. | 1D-CNN | • Achievable data rate.<br>• Packet delivery ratio.<br>• Packet loss rate.<br>• Average end-end-to delay. | V2X communication, in which vehicles on the road share information with everything. |
| [21] | Direct communication between V2X. | N/A | Data-driven | • Vehicle image monitoring.<br>• Driving comfort.<br>• V2X communication. | Communication between vehicles and everything to collect data from various sources to increase driver awareness and decrease collision. |
| [22] | Direct communication between V2X | N/A | CNN | • Preventing collisions and chaos.<br>• Enhancing safety monitoring. | Direct communication between vehicles and everything to collect data from various source. |
| [24] | Direct V2V and communicates with infrastructure through relays. | N/A | LSTM | • System accuracy and real-time recognition. | • V2V communication which allows vehicles to exchange information with other vehicles within their communication range.<br>• Vehicle uses roadside devices for message transmission between vehicle and infrastructure. |
| [25] | Direct communication between AV and infrastructure. | N/A | N/A | • Security<br>• Reliability | Direct communication between AV and infrastructure and between UE and infrastructure. |
| [26] | Direct communication between V2X | N/A | N/A | • Latency<br>• Throughput | Direct communication between vehicles and everything to collect data from various sources. |
| [27] | Direct communication. | N/A | SDN | • Energy efficiency<br>• Latency | From legacy systems, a flexible foundation of 5G automobile services. |
| [28] | Direct V2V communication and communication with I through NodeB. | • Self-learning Robust Resource Allocation approach.<br>• Bernstein Approximation-based Robust Resource Allocation approach | N/A | • System capacity<br>• System QoS | A single cellular vehicle network is thought to exist where VUEs achieve V2V communication. Furthermore, the New Radio Uu interface is used to deliver V2I messages. |
| Proposed model | Communication between AV2X or using any other vehicle to relay the information to any destination. | Lagrange optimization is used in order to maximize energy efficiency and achievable data rate. | 1D-CNN | • Energy efficiency<br>• Achievable data rate | AV2X communication can be relayed or can be direct based on the system requirements. |

## 3. Materials and Methods

This section uses an analytical optimization strategy to describe the suggested method for improving AV performance. Then, a deep neural network design that can be used for AV networks in the real world validates the dataset produced by the analytically suggested model.

### 3.1. System Model and Problem Formulation

It is assumed for the proposed AV network that there are I number of AVs on the road, *C* number of CUE interact with the BS, *D* number of D2D communications, and *V* number of V2V communications; all these transmission links share the same transmitted AV spectrum, as indicated in Figure 1. The goal of this proposed model is to develop an adaptive model to improve the performance of autonomous vehicle communication. Therefore, as shown in Figure 2, the suggested model is able to choose the destination or relay location to communicate with using constrained optimization techniques. Based on the system requirements, the decision between direct communication and cooperative communication between AV2X will be made. Therefore, AVs will communicate directly with the destination if the system need can be satisfied within the transmission distance between AV2X. While AV will use a relay vehicle to connect with the destination if the needed system performance is impacted by the transmission distance between AV2X. Through the use of the Lagrange optimization technique and deep learning algorithm, the suggested model will be assessed in terms of energy efficiency and achievable data rate. According to the proposed model, each autonomous vehicle will choose the best autonomous vehicle to communicate with based on its environment in order to efficiently and reliably obtain specific road information, send information to other autonomous vehicles, or send information to the destination. Thus, the proposed model aims to enhance the communication of the AV network by determining the maximum required distance between the AV and everything, whether it is another AV or relay vehicle, to send the data to the destination.

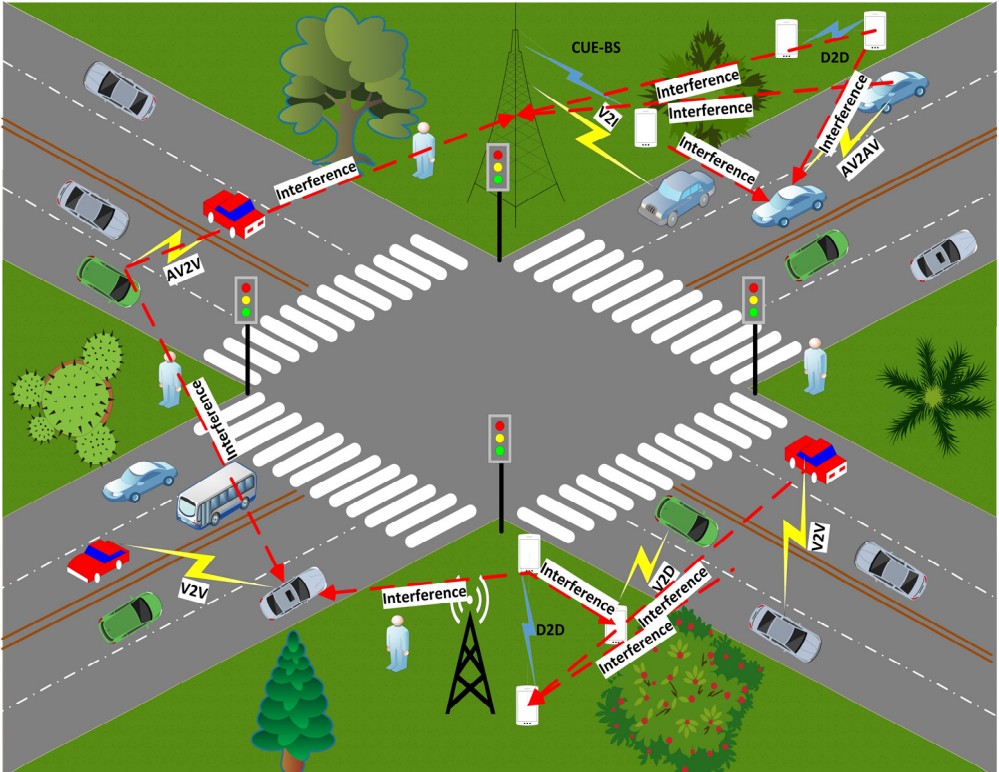

**Figure 1.** Proposed AV system model.

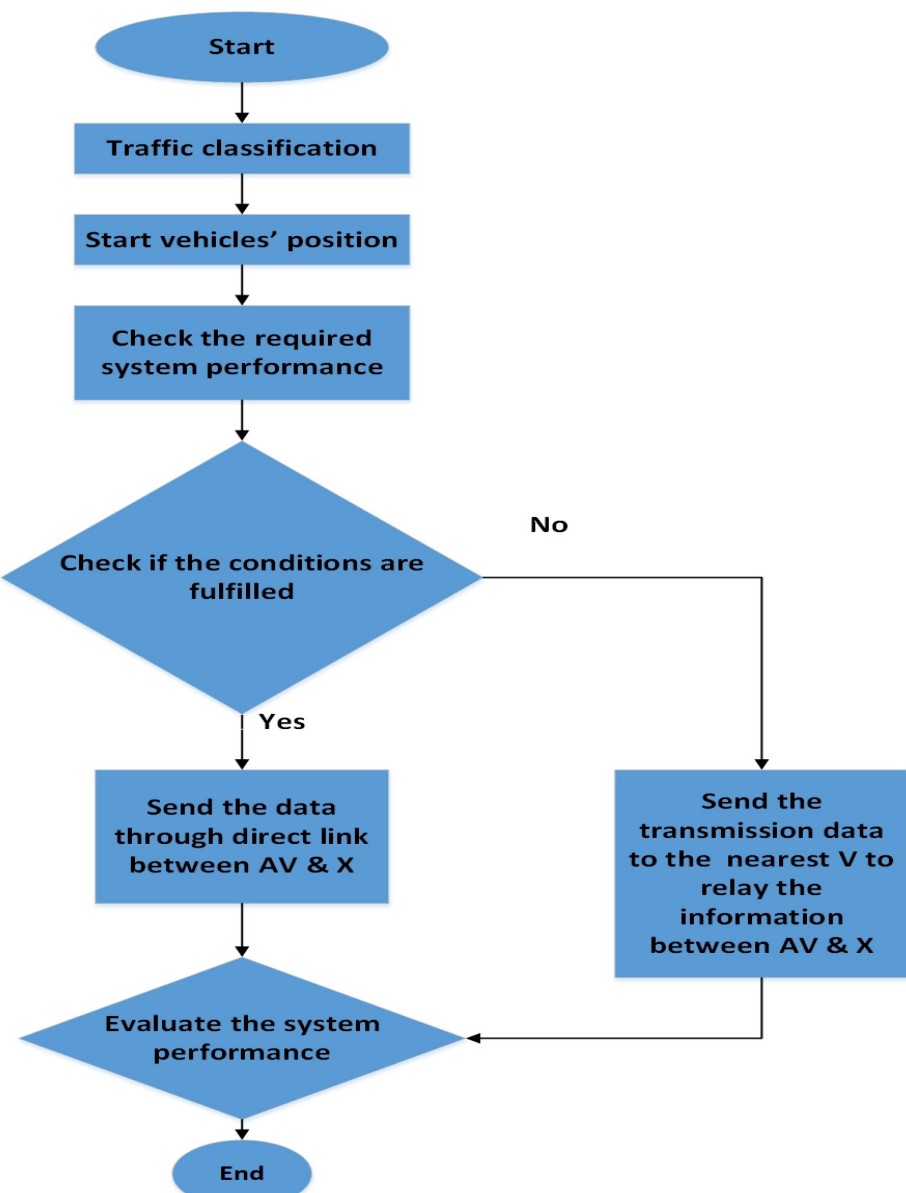

**Figure 2.** Flow chart for the proposed adaptive model.

The following equations illustrate how this results in an optimization of the overall network performance in terms of energy efficiency (*EE*) and the maximum feasible data rate (*R*):

$$
\begin{aligned}
Max \ &\sum_{i=1}^{J} d_{RXi} \\
s.t. \ C1 =: \ &\{I_{2X} \leq \ I_{2Xmax}\} \\
C2 =: \ &\{P_A \leq \ P_{Amax}\}
\end{aligned}
\tag{1}
$$

The transmission distance between relay-vehicle and the required destination, whether it is device, vehicle or even base station, to meet the necessary system performance of the *i*-th path is represented by $d_{RXi}$ in the formulation of the optimization issue. The restriction C1 states that interference ($I_2$) between autonomous vehicle and everything must be less than the maximum permitted interference ($I_{2Xmax}$). $I_{2Xmax}$ can be obtained from

$$
I_{2Xmax} = \sum_{d=1}^{D} P_{D_d max} \ H_{D_d X} + \sum_{c=1}^{C} P_{C_c max} \ H_{C_c X} + \sum_{v=1}^{V} P_{V_v max} \ H_{V_v X}
\tag{2}
$$

where $P_{D_d max}$, $P_{C_c max}$, and $P_{V_v max}$ are the maximum interfered D2D transmission power (Dtx), maximum interfered CUE transmission power, and the maximum interfered V2V transmission power, respectively, which share the same AV spectrum. Symbols $H_{D_d X}$, $H_{C_c X}$, and $H_{V_v X}$ are the channel gain coefficient between transmission D2D device (Dtx) and the destination, transmission CUE and the destination, and transmitted V2V (Vtx) and the destination, respectively. The restriction that the AV transmission power ($P_A$) must be less than the AV transmission power maximum ($P_{Amax}$) is identified by C2.

Rayleigh fading with additive white Gaussian noise (AWGN) and propagation path loss affect the transmission lines between AV2V and AV2X [29]. Statistics-wise, it is believed that the channel fades as various links are independent of one another. The dedicated short-range communications (DSRC) standard is also regarded as the most promising wireless protocol for usage in automobile systems [30]. Short-range wireless communication channels with one-way or two-way functionality are used in DSRC technology and are intended only for usage in automobiles. The MAC and physical layers of the DSRC layer for vehicle communication use the IEEE 802.11p standard. Additionally, the IEEE 802.11p standard is capable of supporting eight distinct data speeds based on the orthogonal frequency division multiplexing (OFDM) technique [31].

### 3.1.1. AV2X Communication Scenario

The AV2X scenario is depicted in Figure 1, in which vehicles go along four streets, each with two sides, and communicate with everything, including infrastructure, other vehicles, and devices; the infrastructure is situated on two sides in the crossroads. As a result, the signal-to-interference-plus-noise ratio between autonomous vehicle-relay-vehicle (AR) and the combined signal-to-interference-plus-noise ratio between autonomous vehicle-everything (AX) and relay-vehicle-everything (RX) can be stated as in [32]:

$$SINR_{AR} = \frac{P_A H_{AR}}{I_1 + N} \tag{3}$$

$$SINR_{AX,RX} = \frac{P_A H_{AX}}{I_2 + N} + \frac{P_A H_{RX}}{I_3 + N} \tag{4}$$

where $H_{AR}$, $H_{AX}$, and $H_{RX}$ are the channel gain coefficient and the transmitted symbol between autonomous vehicle and relay-vehicle, autonomous vehicle and everything, and between relay vehicle and everything, respectively. $I_1$, $I_2$, and $I_3$ represent the interference that occurs between autonomous vehicle and relay-vehicle link, autonomous vehicle and everything link, and between relay vehicle and everything link, respectively. Symbol $N$ represents the noise power. Therefore, $d_{RX}$ can be deduced from Equation (4) as follows:

$$d_{RX} = \left[ \frac{((I_2 + N)SINR_{AR,AX} - P_A H_{AX})P_{Lo}(I_3 + N)}{P_A(I_2 + N)} \right]^{-\frac{1}{\alpha}} \tag{5}$$

where the parameters $P_{Lo}$ and $\alpha$ are the path loss constant and the path loss exponent for the transmission links. Equation (5) will be used to optimize AV energy efficiency (EE) and achievable data rate (R), which are expressed as follows:

$$EE = \frac{R_{RX}}{P_A + P_o} + \frac{R_{AX}}{P_A + P_o} \tag{6}$$

$$R = R_{RX} + R_{AX} \tag{7}$$

where $R_{RX}$ and $R_{AX}$ are the achievable data rate between relay-vehicle and everything and between autonomous vehicle and everything. $P_A$ and $P_o$ represent autonomous vehicle

transmission power and internal circuitry power, respectively. Thus, $R_{RX}$ and $R_{AX}$ can be given as

$$R_{RX} = B log_2 \left( 1 + \frac{P_A H_{RX}}{\sum_{d=1}^{D} P_{D_d} H_{D_d X} + \sum_{c=1}^{C} P_{C_c} H_{C_c X} + \sum_{v=1}^{V} P_{V_v} H_{V_v X} + N} \right) \quad (8)$$

$$R_{AX} = B log_2 \left( 1 + \frac{P_A H_{AX}}{\sum_{d=1}^{D} P_{D_d} H_{D_d X} + \sum_{c=1}^{C} P_{C_c} H_{C_c X} + \sum_{v=1}^{V} P_{V_v} H_{V_v X} + N} \right) \quad (9)$$

Symbols $P_{D_d}$, $P_{C_c}$ and $P_{V_v}$ are the interfere transmission device (Dtx) transmission power, interfere transmission CUE transmission power, and interfere transmission vehicle (Vtx) transmission power, respectively. $H_{D_d X}$, $H_{C_c X}$, and $H_{V_v X}$ are the channel gain coefficient between interfere transmission device (Dtx) and everything, interfere transmission CUE and everything, and interfere transmission vehicle (Vtx) and everything, respectively.

As demonstrated in Equation (1), the major goal of the proposed approach is to maximize total energy efficiency (*EE*) and feasible data rate (*R*) under varied environmental conditions. As a result, the Lagrangian of the optimization problem stated in (1) is as follows:

$$L\{d_{AR}, I_2, P_A, \lambda_1, \lambda_2\} = d_{RX} - \lambda_1 (I_{2max} - I_2) - \lambda_2 (P_{Dmax} - P_D) \quad (10)$$

where the non-negative Lagrangian multipliers are $\lambda_1$ and $\lambda_2$. It is possible to determine the value of $\lambda_1$ and $\lambda_2$ for satisfying the constraint of the optimization problem for ($d_{AR}$) by considering the derivative of Equation (10) with respect to $I_2$ and $P_A$. Thus, $\lambda_1$ and $\lambda_2$ can be determined as

$$\lambda_1 = \frac{1}{\alpha} \left[ \frac{P_{Lo}(I_3 + N) SINR_{RX,AX}}{P_A{}^2} \right] \left[ \frac{((I_2 + N) SINR_{RX,AX} - P_A H_{AX}) P_{Lo}(I_3 + N)}{P_A(I_2 + N)} \right]^{-\frac{1}{\alpha} - 1} \quad (11)$$

$$\lambda_2 = \frac{1}{\alpha} \left[ \frac{H_{AX} P_{Lo}(I_3 + N)}{(I_2 + N)^2} \right] \left[ \frac{((I_2 + N) SINR_{RX,AX} - P_A) P_{Lo}(I_3 + N)}{P_A(I_2 + N)} \right]^{-\frac{1}{\alpha} - 1} \quad (12)$$

Additionally, the optimal required interference between AV and everything ($I_2$) and the optimal required AV transmission power ($P_A$) can be obtained by taking the derivative of Equation (10) with respect to $\lambda_1$, $\lambda_2$, and $\lambda_3$ as follows:

$$I_2 = I_{2max} \quad (13)$$

$$I_2 = \sum_{d=1}^{D} P_{D_d max} H_{D_d X} + \sum_{c=1}^{C} P_{C_c max} H_{C_c X} + \sum_{v=1}^{V} P_{V_v max} H_{V_v X} \quad (14)$$

$$P_A = P_{Amax} \quad (15)$$

### 3.2. Dataset Generation

To produce the required datasets, the equations of the proposed model outlined in Sections 3.1 and 3.1.1 were implemented using MATLAB simulations. The values of the simulation parameters are shown in Table 2. The datasets will be used to train models that will be put on all transmitting devices to prevent interference from occurring during any medical reception. There are 20,311 records, each record representing a unique combination of the distances between AX and everything ($d_{AX}$), the required signal-to-interference-plus-noise-ratio threshold ($SINR_{th}$), the AV transmission power ($P_A$), CUE transmission power ($P_C$), D2D transmission power ($P_D$), and V2V transmission power ($P_V$).

**Table 2.** System simulation parameters.

| Parameter | Value |
|---|---|
| $N$ | $-174$ dBm/Hz [4,33] |
| $B$ | 10 Mbit/s [4,34] |
| $\alpha$ | 4 |
| $P_A$ | 23 dBm [4,35] |
| $P_C$ | 23 dBm [4,35] |
| $P_D$ | 23 dBm [4,35] |
| $P_V$ | 23 dBm [4,35] |
| $SINR_{th}$ | 20 dB [4] |
| Path loss between AV and everything | $(128.1 + 37.6 \, log_2(d_{AX}/km))$ |
| Path loss between CUE and BS | $(148 + 40 \, log_2(d_{CB}/km))$ |
| Path loss between D2D links | $(128.1 + 37.6 \, log_2(d_{DD}/km))$ |
| Path loss between V2V links | $(128.1 + 37.6 \, log_2(d_{VV}/km))$ |

The Pearson coefficients in Figure 3 show the link between each input and output parameter. The graph shows that the output $d_{RX}$, *EE*, and *R* have a low correlation with $P_C$, $P_D$, and $P_V$ parameters. Also, parameter R has a strong positive correlation with the $P_A$ parameter. The deep learning model must learn the impact of each of these parameters, and the effect of the correlation presented will be explained in the results section.

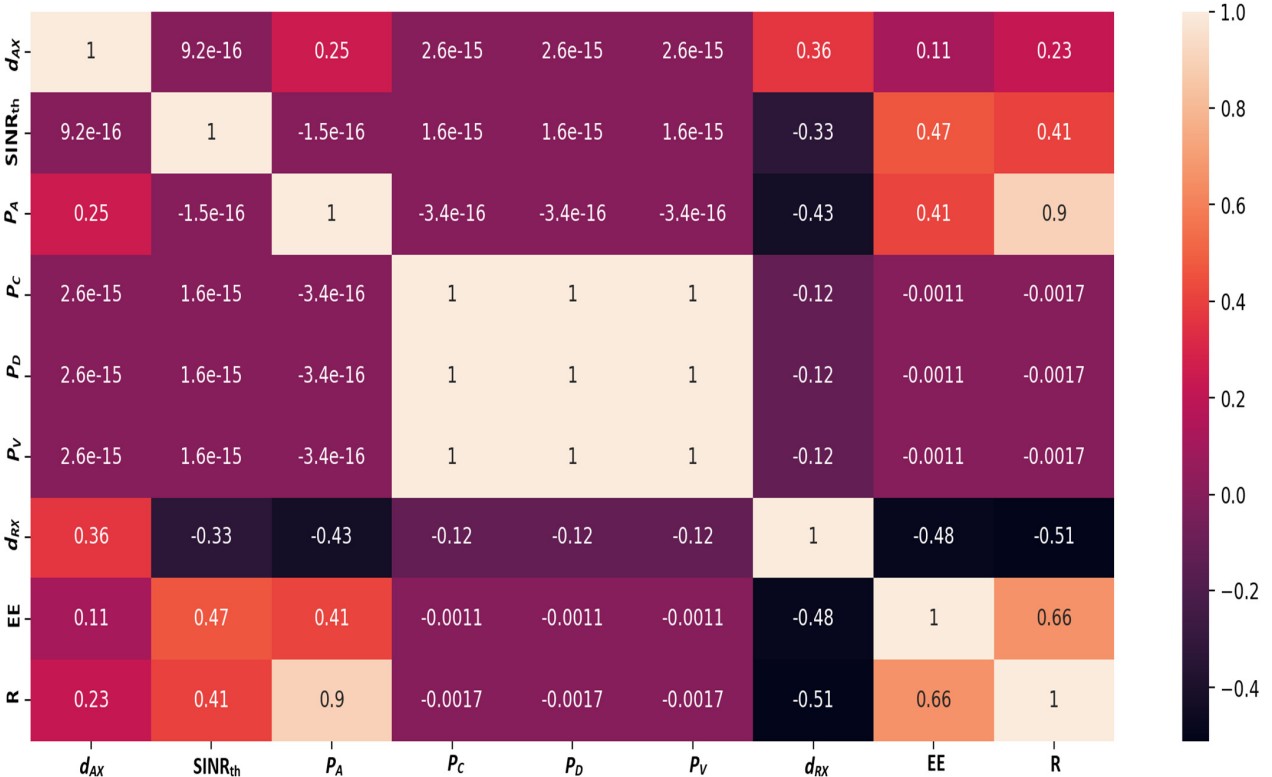

**Figure 3.** Pearson correlation coefficients of each input parameter ($d_{AX}$, $SINR_{th}$, $P_A$, $P_C$, $P_D$, and $P_V$) and the output ($d_{RX}$, *EE*, and *R*).

### 3.3. Proposed Deep Learning Model

This section proposes and explains the proposed deep learning model. Prior to inputting the variables into the recommended deep learning model, a normalization step is required to make the learning stage of the model weights easier. The variables are all normalized before being entered into the model using the min-max scaling strategy. The last dense layer's output parameters $d_{RX}$, *EE*, and *R* are calculated using the five input variables $d_{AX}$, SINRth, $P_A$, $P_C$, $P_D$, and $P_V$. As shown in Figure 4, the model is made up of three distinct phases: 1D-CNN, flattening, and thick layers. Three 1D-CNN layers are

employed to process the normalized input parameters, each with 128 filters and a kernel of size 1.

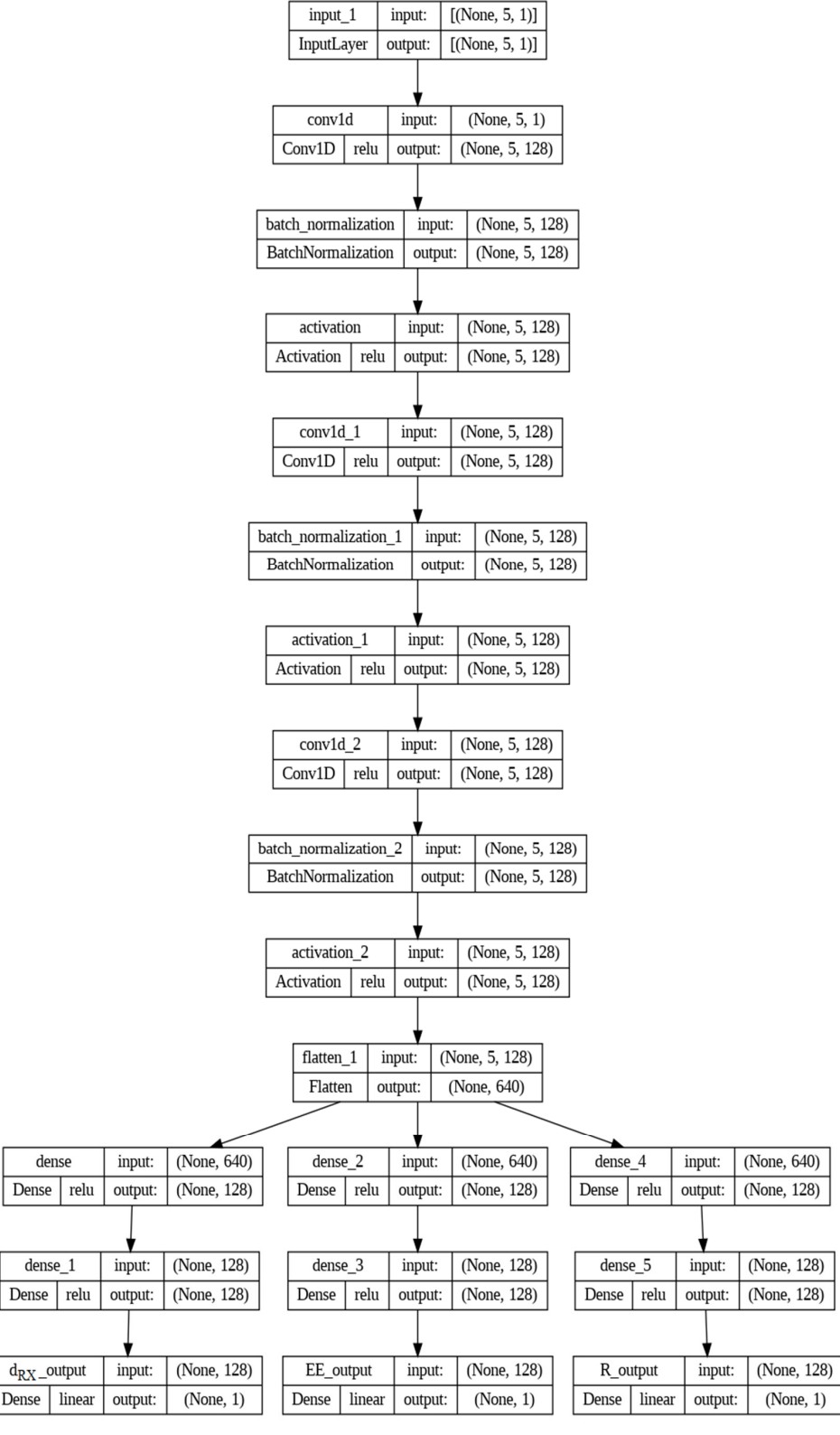

**Figure 4.** Proposed deep learning model.

Each 1D-CNN layer generates padded results in order to maintain the width of the output matrix consistently. The output of the third 1D-CNN is then delivered into a flattening layer, which reformats the dimension and prepares it for feeding into the dense

layers. Following the flattening layer are six dense layers that yield the regression result. Before settling on the number of filters for the 1D-CNN and the number of nodes for the dense layers, a grid search was performed to evaluate numerous combinations. All hidden layers were activated using the Rectified Linear Unit (ReLU). The grid search also included experimenting with various strategies that might follow the hidden layers in the suggested model while taking activation function selection into account. The output of each hidden layer was fed into the best-performing parametric rectified linear unit (PReLU) activation function. The optimization used in the proposed model with the mean absolute error (MAE) loss function and root mean squared error (RMSE) as targets is the adaptive moment (Adam), which may adaptively learn the appropriate parameters based on the learning process. MAE is responsible for assessing the average difference between real and predicted values, whereas RMSE is the root square of the average of the squared differences between actual and predicted values, as represented by

$$MAE = \frac{\sum_{j=1}^{n} |y_j - x_j|}{n} \tag{16}$$

$$RMSE = \frac{\sum_{j=1}^{n} (y_j - x_j)^2}{n} \tag{17}$$

where n is the total number of recorded data, $y_j$ denotes the actual value, and $x_j$ denotes the predicted value. The experiments utilized to train, test, and evaluate the proposed model are described in the next section.

## 4. Results and Discussion

The performance of the proposed analytical and deep learning models is presented in this section. Furthermore, the efficacy of the proposed approach was tested using MATLAB and Python simulations in terms of optimized energy efficiency and achievable data rate.

Figure 5 depicts the testing and evaluation of the proposed deep learning model described in Section 3.3. The datasets were divided into two parts: 80% training data and 20% test data. Figure 5a–c indicate the training and validation mean absolute error for the needed $d_{RX}$, $EE$, and $R$, respectively; all of the figures illustrate that there was no need for extra training beyond epoch 100 because the outcomes were barely changing. Furthermore, Figure 5d shows that the independent training and validation error losses decrease and are maintained at a specific point, as well as that the independent training and validation errors for each output were close to the same values, indicating that the proposed model was not overfitted or underfitted.

Figure 6 shows the required $SINR_{th}$ versus the overall achievable data rate for the analytical and deep learning models and compared with the model proposed in [4]. To evaluate the effectiveness of the proposed model, high interference is assumed where all the interfere devices send their data with the highest transmission power. Thus, it can be observed from Figure 6 that the proposed model outperforms the model proposed in [4] in terms of achievable data rate for both the analytical and deep learning models. Also, it can be mentioned for both the analytical and deep learning models that the achievable data rate increases with the increase of $SINR_{th}$. In the context of V2V communication, obtaining a higher data rate means that vehicles can exchange information more quickly and effectively. This results in faster and more reliable communication between vehicles, allowing them to transmit real-time information such as traffic conditions, road hazards, or changes in driving behaviors. A higher achievable data rate in V2V communication enables more efficient and rapid exchanges between autonomous vehicles, thereby contributing to safer and smoother traffic flow.

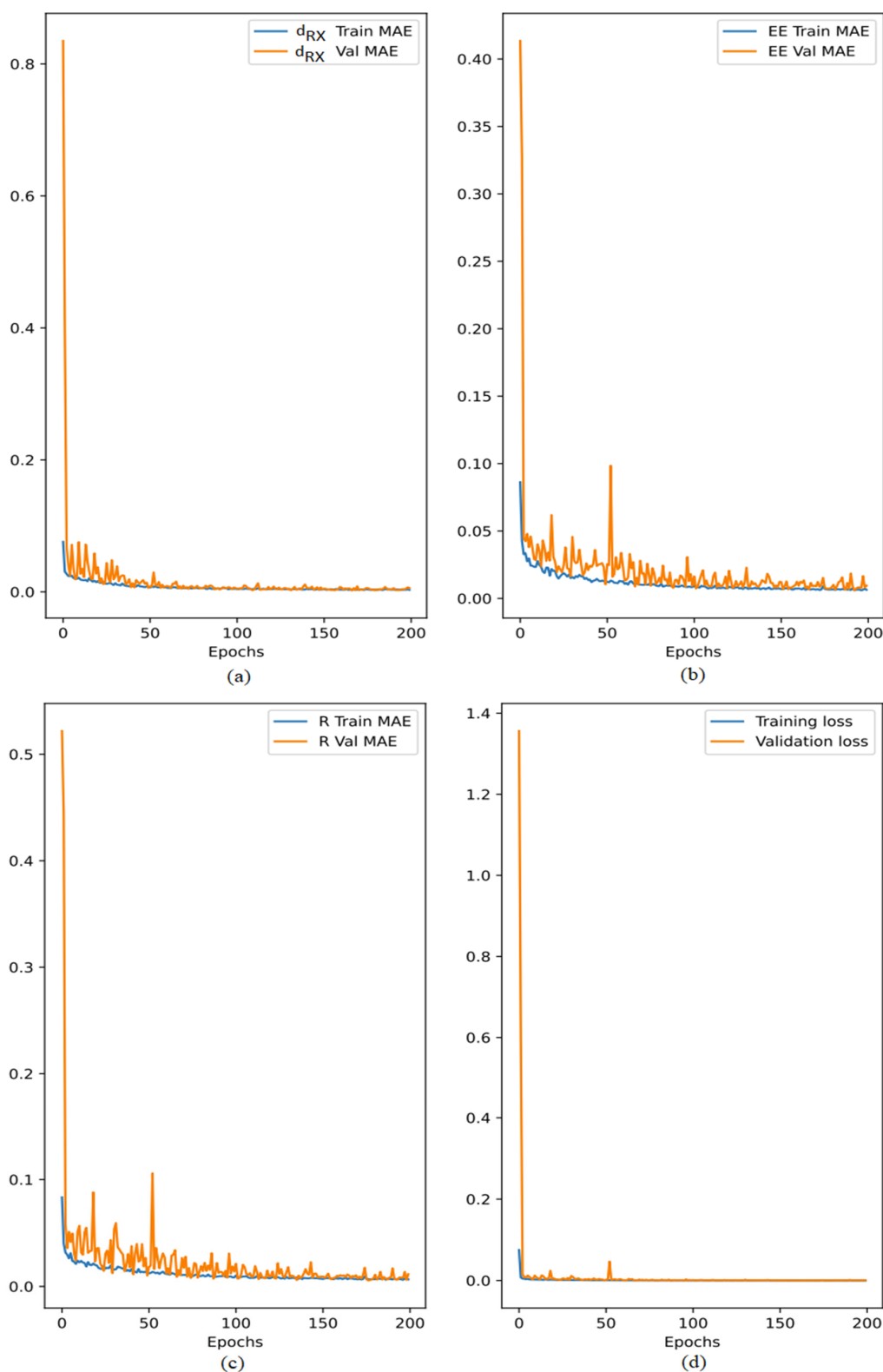

**Figure 5.** Training and validation mean absolute error generated during training of the proposed model. (**a**) $d_{RX}$ Training and validation mean absolute error vs epochs (**b**) *EE* Training and validation mean absolute error vs epochs (**c**) *R* Training and validation mean absolute error vs epochs (**d**) Training and validation mean absolute error vs epochs.

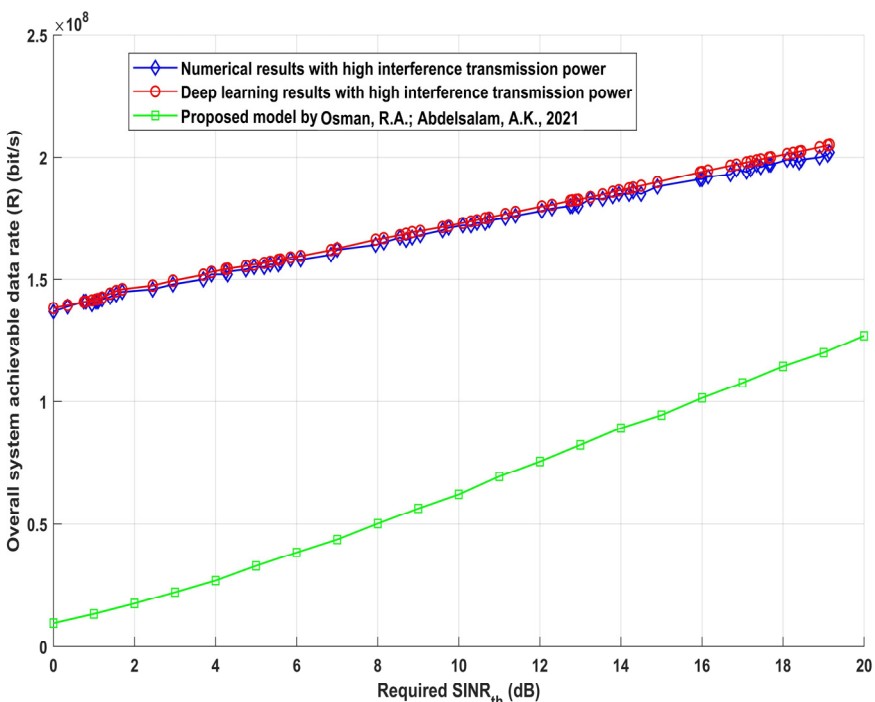

**Figure 6.** Required $SINR_{th}$ ($d_{IR}$) vs. overall system achievable data rate ($R$) [4].

Furthermore, the system was evaluated in terms of overall system achievable data rate vs. AV transmission power, as depicted in Figure 7. In Figure 7, low interference is assumed where all the interfere devices sent data with low transmission power (17 dBm). The system is evaluated for two different $SINR_{th}$, as demonstrated in Figure 7a,b. Figure 7a depicts that when $SINR_{th}$ is 0 dB while the system is facing low interference, the proposed model performs well when $P_A$ exceeds 7 dBm for both the analytical and deep learning models. Additionally, the same result is obtained when $SINR_{th}$ is 20 dB, as shown in Figure 7b; the system outperforms the model proposed in [4] when $P_A$ exceeds 7 dBm for the analytical and deep learning models. It is worth mentioning that increasing $P_A$ increases the overall system achievable data rate by helping to overcome interference and improve system performance. Increasing transmission power above a specific threshold, as specified in Figure 7 (7 dBm), is advantageous because it enhances the sent signal. This increase in signal strength aids in overcoming interference, improving the signal-to-noise ratio, and increasing the overall system's data rate. This boost in transmission power benefits both the proposed analytical and deep learning models, resulting in improved system performance even in low-interference conditions. The results obtained show the effectiveness of the proposed model as it proves its ability to adapt itself under different circumstances and environmental conditions to face the challenges of achieving the required system performance and sending reliable data to any destination.

When evaluating the system's performance for the same previously assumed scenario, assuming high interference power where all interfere devices send data with high transmission power (23 dBm), the same result is reached in terms of achievable data rate ($R$), as shown in Figure 8. Figure 8a shows that when $SINR_{th}$ is 0 dB and the system is subjected to significant interference, the suggested model works effectively when $P_A$ exceeds 8 dBm for both the analytical and deep learning models. Furthermore, when $SINR_{th}$ is 20 dB, as shown in Figure 8b, the system outperforms the model given in [4] when $P_A$ exceeds 8 dBm for both the analytical and deep learning models. As discussed previously, boosting $P_A$ raises the overall system achievable data rate since it helps overcome interference and improve system performance. The results obtained in Figure 8 show the capacity of the proposed model to learn and adapt to complex interference settings, capture non-linear correlations, and efficiently exploit available data for interference mitigation. These benefits

enable the model to outperform standard analytical models, particularly in circumstances involving heavy interference and variable $SINR_{th}$.

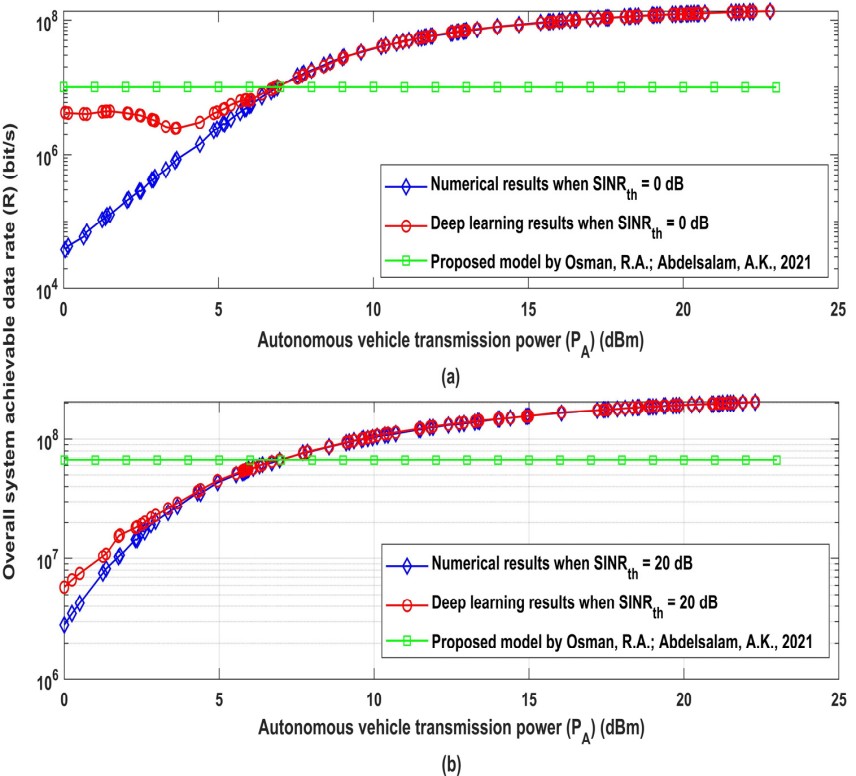

**Figure 7.** Autonomous vehicle transmission power ($P_A$) and overall system achievable data ($R$) (low interference). (**a**) Low required $SINR_{th}$ (**b**) High required $SINR_{th}$ [4].

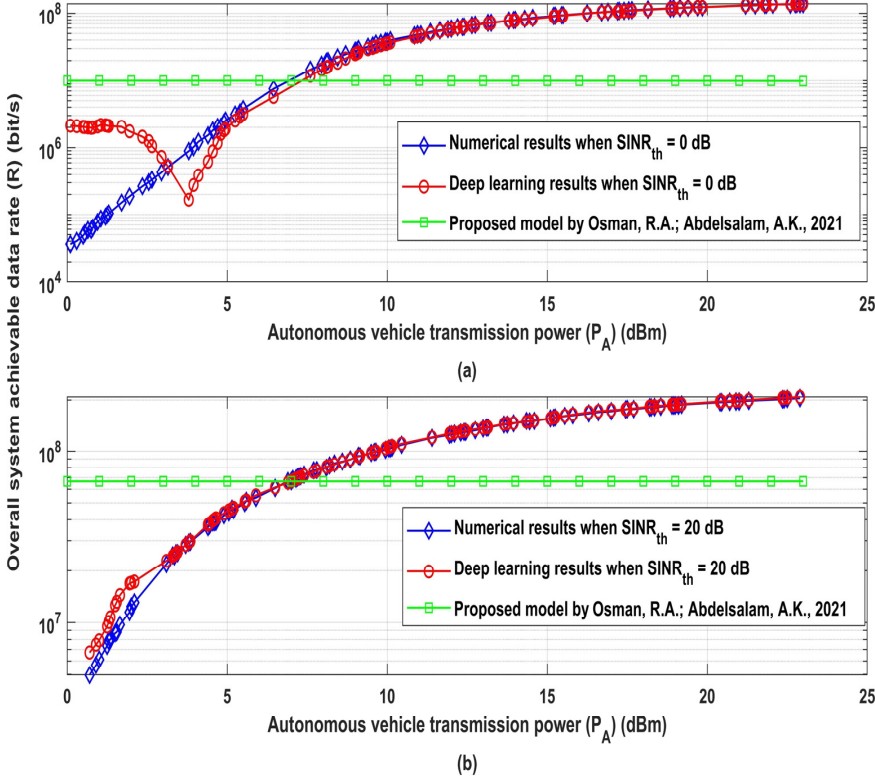

**Figure 8.** Autonomous vehicle transmission power ($P_A$) and overall system achievable data ($R$) (high interference). (**a**) Low required $SINR_{th}$ (**b**) High required $SINR_{th}$ [4].

For further evaluation, we have compared the performance of the proposed approach during high and low interference for two different $SINR_{th}$, which are 0 dB and 20 dB, as illustrated in Figure 9. Figure 9a demonstrates the system's achievable data rate when the system faces low interference; thus, it can be found that the achievable data rate increases when $P_A$ increases for both assumed $SINR_{th}$. Additionally, it is worth mentioning that the proposed approach performs well when $SINR_{th}$ is 20 dB compared with the performance when $SINR_{th}$ is 0 dB. The same performance is obtained when the system faces high interference, as depicted in Figure 9b; it is noted that the achievable data rate increases when $P_A$ increases for both assumed $SINR_{th}$, and the system performs well when $SINR_{th}$ is 20 dB compared with 0 dB for both the analytical and deep learning models. From the results obtained, it should be mentioned that while increasing the needed $SINR$ can possibly result in higher achievable data rates, there is a practical limit. At some point, a careful balance must be struck between increasing transmission power to achieve higher $SINR$ requirements and the risk of increased interference from neighboring devices or other environmental conditions. Moreover, raising the required $SINR$ threshold necessitates a stronger and more robust sent signal, resulting in greater interference mitigation, decreased errors, increased channel capacity, and more efficient modulation and coding. All of these elements contribute to a higher achievable data rate, which shows the significance of the proposed approach.

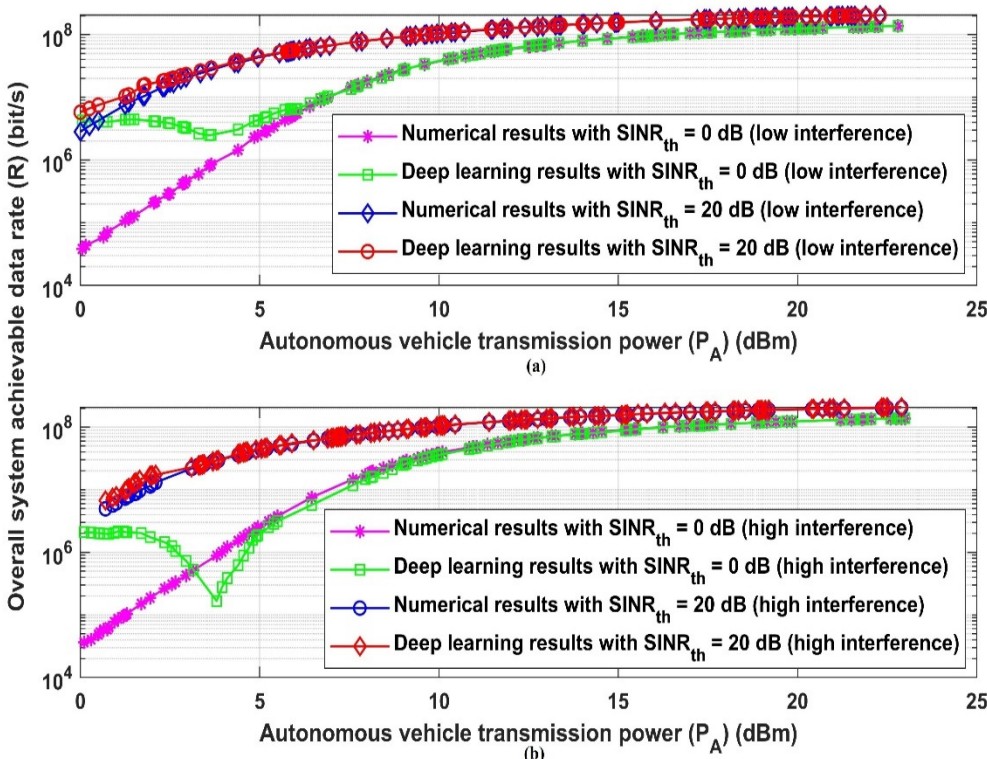

**Figure 9.** Autonomous vehicle transmission power ($P_A$) and overall system achievable data (R). (**a**) Comparison between low and high SINR$_{th}$ in case of low interference (**b**) Comparison between low and high SINR$_{th}$ in case of high interference.

Furthermore, the system's performance has been evaluated in terms of *EE* vs. the required $SINR_{th}$ when the system faces low and high interference, as shown in Figure 10. Figure 10 shows the proposed approach can overcome interference whether low or high and reach the optimum *EE* for each $SINR_{th}$. Also, it can be noted that increasing the required $SINR_{th}$ leads to increasing the overall system *EE* for both analytical and deep learning models. Higher *EE* shows that the system is making better use of its energy resources to meet communication goals. Additionally, increasing the needed $SINR_{th}$ improves energy efficiency due to system modifications such as transmission power optimization, enhanced

signal quality, adaptive modulation and coding, and dynamic resource allocation. These changes contribute to conveying data more effectively per unit of energy spent, resulting in the observed rise in overall system *EE* for both the analytical and deep learning models.

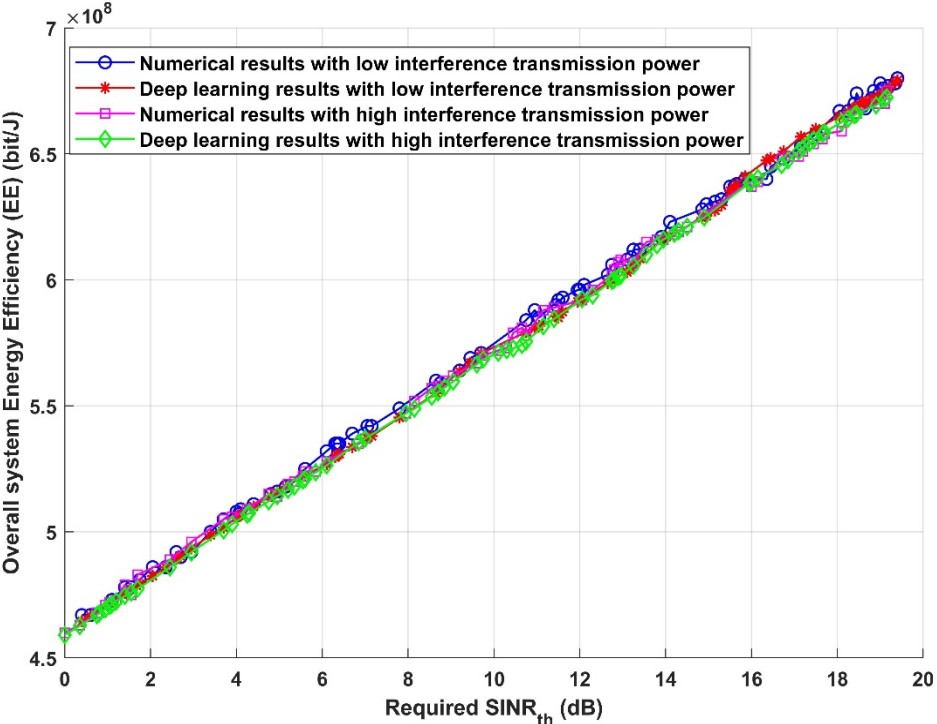

**Figure 10.** Required $SINR_{th}$ ($d_{IR}$) vs. overall system energy efficiency (*EE*).

For further investigation, the system is evaluated in terms of *EE* and versus variable AV transmission power ($P_A$), as well as when the system encounters high and low interference for two distinct $SINR_{th}$ values of 0 dB and 20 dB, as shown in Figure 11. Figure 11a shows the system *EE* when the system is subjected to low interference; consequently, *EE* increases as $P_A$ increases for both assumed $SINR_{th}$. Furthermore, when SINR$_{th}$ is 20 dB, the suggested approach performs well in terms of *EE* when compared to the performance when *SINRth* is 0 dB. When the system is subjected to severe interference, as shown in Figure 11b, it is observed that *EE* increases with the increment of $P_A$ for both assumed $SINR_{th}$, and the system works well when $SINR_{th}$ is 20 dB compared to 0 dB for both the analytical and deep learning models. Thus, it is worth mentioning that increasing transmission power enhances the transmitted signal, boosting *SINR* and resulting in improved *EE* by delivering more data per unit of energy. Also, increasing the required $SINR_{th}$ needs a stronger signal, resulting in increased energy efficiency by focusing energy on sending a reliable signal even in the presence of interference and noise.

To emphasize the significance of the suggested model, a comparison with one of the most current research papers [28] was conducted, as shown in Figure 12. The findings shown in Figure 12 clearly show that our proposed approach outperforms the model introduced in [28] in terms of achievable data rate. This superiority can be due to the adaptive nature of our proposed model, which allows it to dynamically respond to channel characteristics such as transmission distance, path loss, and interference power. The proposed model consistently achieves the desired system performance by picking the most appropriate transmission scheme based on these conditions. Additionally, it should be noted that when the transmission power reaches 30 dBm, the model proposed in [28] outperforms our proposed model. However, this is noteworthy as our proposed model does not aim to reach such high transmission power values. Instead, it adapts itself based on channel conditions to select the suitable power level that fulfills the system requirements, ultimately maximizing the system's energy efficiency.

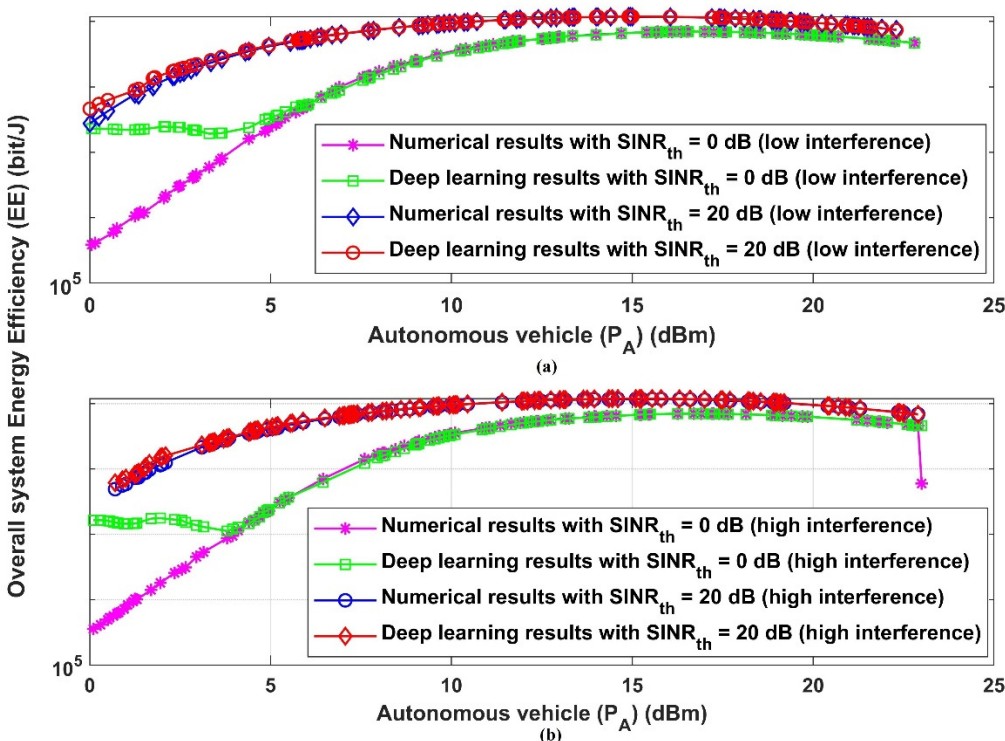

**Figure 11.** Required *SINR*$_{th}$ (*d*$_{IR}$) vs. overall system energy efficiency (*EE*). (**a**) *EE* performance with low and high SINR$_{th}$ in case of low interference (**b**) *EE* performance with low and high SINR$_{th}$ in case of high interference.

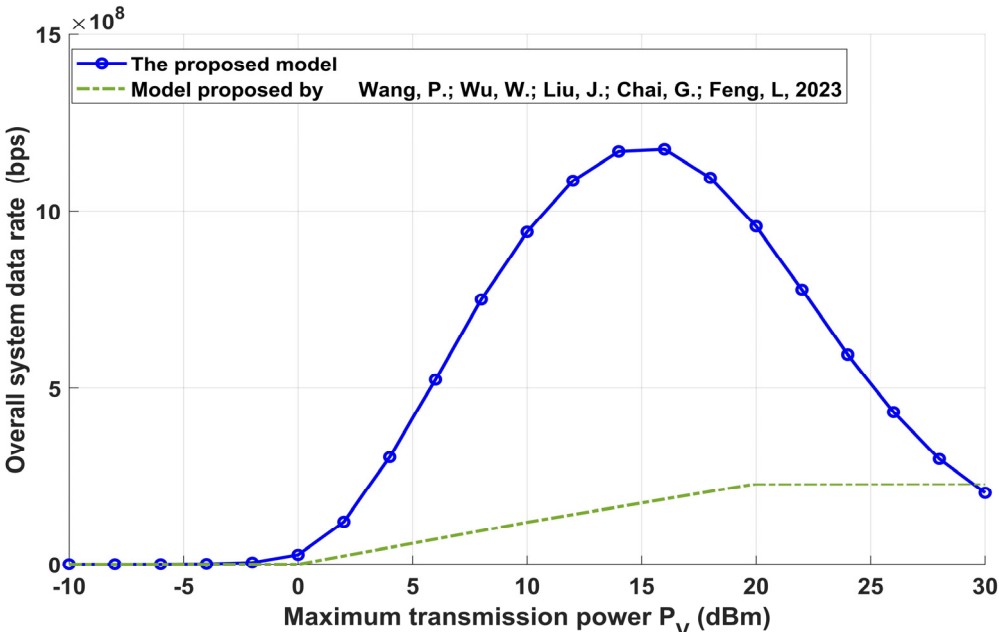

**Figure 12.** Comparison of achievable data rate: proposed model vs. [28].

Finally, it can be concluded from the results obtained that the deep learning model and the optimization proposed have the adaptability and capabilities to react to varied system environmental condition and then improving the system's performance in terms of energy efficiency and achievable data rate. Additionally, it is worth mentioning that despite the substantial contributions offered by the proposed model to improving vehicular network connectivity through our suggested adaptive AV2X model, it is vital to recognize the proposed model's limitations. One notable limitation pertaining to the scope of the

investigated model is coding and modulation. Modulation methods and coding techniques are important components of wireless communication systems; they were purposefully left out of our inquiry model. The proposed model's aims were customized to solve specific difficulties connected to fulfilling the system requirements by dynamically determining the optimum transmission distance with the suitable transmission power based on real-time system performance for enhancing energy efficiency and achievable data rate. Modulation and coding are crucial parts of autonomous vehicle communication, and future research efforts may dive deeper into these areas.

## 5. Conclusions

In this paper, firstly, a novel adaptive AV-to-everything communication approach is proposed with implementation of an optimization technique and a distributed deep learning model in order to enhance the connectivity of vehicular communication. Then, the problem of ensuring a reliable communication between AV and everything is discussed under various environmental conditions. The required inter-vehicle position is initially determined using the Lagrange optimization technique, then simulated using MATLAB. Based on the data provided by MATLAB, the 1D-CNN-based deep learning model was suggested. The goal of employing 1D-CNN is to have lowest computational complexity, which allows for processing energy efficiency and overall achievable data rate, making it perfect for usage in real-time applications. As a result, the deep learning model on AVs was able to determine the correct inter-vehicle location in order to reach a nearly optimal result. Using both techniques, the ideal inter-vehicle position between the AV and any destination, whether it was another vehicle, device, or base station, was predicted. As a result, in order to meet the system's performance requirements, the analytics of the results of forecasting the optimal required inter-vehicle location between the AV and everything were then displayed. Following that, based on the findings obtained in terms of system energy efficiency and achievable data rate, it was discovered that the proposed model can demonstrate the greatest performance under various environmental situations. Furthermore, utilizing analytical and deep learning techniques, it has been proven that the inter-vehicle location can alter depending on a variety of environmental factors. The results reveal that boosting $P_A$ increases energy efficiency and achievable data rate since it helps to overcome any type of interference, whether high or low. Furthermore, the effect of $SINR_{th}$ on the energy efficiency and achievable data rate was investigated. As a result, it has been proven that increasing the SINRth improves overall energy efficiency and achievable data rate for both types of assumed interferences. Finally, the findings reveal that the suggested model may deliver the desired AV performance while maintaining an acceptable level of system efficiency and reliability. Furthermore, another important direction for future work is to assess the effectiveness of the proposed resource allocation technique using experimental data from a hardware implementation or other deep learning techniques.

**Funding:** This research received no external funding.

**Data Availability Statement:** Not applicable.

**Conflicts of Interest:** The author declares no conflict of interest.

## Glossary

List of symbols

| | |
|---|---|
| $B$ | System bandwidth |
| $I_{2X}$ | Interference at any destination |
| $I_{2Xmax}$ | The maximum permitted interference |
| $C1, C2$ | Lagrangian optimization constraints |

| | |
|---|---|
| $d_{RXi}$ | The transmission distance between relay-vehicle and the required destination between everything of the *i*-th |
| $P_{D_d max}$ | The maximum interfered D2D transmission power (Dtx). |
| $P_{C_c max}$ | The maximum interfered CUE transmission power |
| $P_{V_v max}$ | The maximum interfered V2V transmission power |
| $H_{D_d X}$ | The channel gain coefficient between transmission D2D device (Dtx) and the destination |
| $H_{C_c X}$ | The channel gain coefficient between CUE and the destination |
| $H_{V_v X}$ | The channel gain coefficient between transmitted V2V (Vtx) and the destination |
| $P_A$ | Autonomous vehicle transmission |
| $P_{Amax}$ | Maximum autonomous vehicle transmission |
| $SINR_{AR}$ | Signal-to-interference-plus-noise ratio between autonomous vehicle-relay-vehicle (AR) |
| $SINR_{RX,AX}$ | The combined signal-to-interference-plus-noise ratio between autonomous vehicle-everything (AX) and relay-vehicle-everything (RX) |
| $H_{AR}$ | The channel gain between autonomous vehicle and relay-vehicle |
| $H_{AX}$ | The channel gain between autonomous vehicle and everything |
| $H_{RX}$ | The channel gain between relay vehicle and everything |
| $I_1$ | The interference that occurs between autonomous vehicle and relay-vehicle link |
| $I_2$ | The interference that occurs between autonomous vehicle and everything link |
| $I_3$ | The interference that occurs between relay vehicle and everything link |
| $N$ | The noise power |
| $R_{RX}$ | The achievable data rate between relay-vehicle and everything |
| $R_{AX}$ | The achievable data rate between autonomous vehicle and everything |
| $P_A$ | Autonomous vehicle transmission power |
| $P_o$ | Internal circuitry power |
| $EE$ | System energy efficiency |
| $R$ | Overall achievable data rate |
| $P_{D_d}$ | The interfere transmission device (Dtx) transmission power |
| $P_{C_c}$ | The interfere transmission CUE transmission power |
| $P_{V_v}$ | The interfere transmission vehicle (Vtx) transmission power |
| $H_{D_d X}$ | The channel gain coefficient between interfere transmission device (Dtx) and everything |
| $H_{C_c X}$ | The channel gain coefficient interfere transmission CUE and everything |
| $H_{V_v X}$ | The channel gain coefficient between interfere transmission vehicle (Vtx) and everything |
| $PLo$ | The path loss constant |
| $SINRth$ | The required signal-to-interference-plus-noise-ratio threshold |
| $n$ | The total number of recorded data |
| $y_j$ | The actual value |
| $x_j$ | The predicted value |
| $\lambda_1, \lambda_2$ | non-negative Lagrangian multipliers |
| $\alpha$ | Path loss exponent |

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
