# Peer review of "Optimizing Autonomous Vehicle Communication through an Adaptive Vehicle-to-Everything (AV2X) Model: A Distributed Deep Learning Approach"

_electronics, doi:10.3390/electronics12194023_

Round 1
Reviewer 1 Report
1. The second part of the paper needs to lack a summary of the previous references, as well as a clear explanation of how the methodology proposed in this paper effectively solves the problems of the previous studies.
2. In the Abstract and Introduction, the authors repeatedly refer to "optimization algorithms", however, it is not clear which optimization algorithms are being referred to. Is it possible to specify the specific optimization algorithms used, rather than just mentioning them in general terms?
3. The paper omits specific definitions and symbolic descriptions of some of the parameters and variables used in the methodology section, which makes some of the formulas difficult to understand, and it is recommended that they be checked and additional descriptions be provided.
4. Formulas should be laid out as neatly as possible, and the headings of tables should be formatted, wherever possible, in accordance with the template, with appropriate modifications.
5. The names of figures 6-8, such as titles and legends, should not be exactly the same, and it is suggested that appropriate changes be made to show more clearly the differences between each figure.
6. It is recommended that flow charts, etc., be added to the methodology section in order to provide a more intuitive and clearer presentation of the principles and processes of the proposed model.
7. For the value of the parameter α, the value of 3 in the literature [4] is not identical to the value of 4 in this paper. Please consider whether further specification of this parameter is desirable.
8. It is recommended that the results and discussion section provide a fuller and more in-depth analysis and interpretation of the data obtained, as well as a more comprehensive and objective comparison and assessment with the previous methodology, highlighting the strengths and limitations of the proposed model.
Requires minor editing of English.
Reviewer 2 Report
This paper aims to present a novel model designed to predict and enhance vehicle communications within the future V2X context. The topic is undoubtedly intriguing, nevertheless, there are some evident shortcomings in this paper that necessitate the author's attention and correction prior to resubmission.
- The communication scheme employed to extract data from the 0 dB SINR channel in this model remains unclear. I assume that a processing gain is incorporated during OFDM signal demodulation. Could you please provide more details on how your model achieves a 0 dB SINR and the associated demodulation process?
- In Figure 6, there appears to be an improvement in data rate with higher transmission power, but the SINR remains constant. Does this imply that the interference signal is also increasing concurrently? How is the data rate enhanced while maintaining a 10MHz signal bandwidth under varying SNR conditions? This seems to deviate from fundamental theory. Is there a change in bandwidth or an expectation of adaptive modulation formats?
- Figure 5 displays a maximum data rate exceeding 200Mbps for a signal with 10MHz bandwidth and a spectral efficiency of 1b/s/Hz. Could you elucidate how this data rate is achieved? Similar to the previous question, are high-order modulation formats expected?
- Could you please specify the achieved bit error rate (BER) corresponding to various SINR levels?
- One of the objectives mentioned in the introduction is to predict the maximum permissible transmission distance. However, I couldn't find any results related to transmission distance in the paper. Could you clarify this discrepancy?
- The path loss and data rate models seem rather simplistic, and real-life environments can significantly impact these models, potentially leading to deviations from real-world scenarios. Can you elaborate on how your results may differ from actual situations due to environmental factors?
- Figure 7 depicts a nearly linear relationship between SINR and Energy Efficiency, which appears somewhat optimistic. Could you provide a more detailed explanation of how this relationship is derived and why it might be realistic in practical scenarios?
- The paper lacks a quantitative comparison with other models, aside from your previous results. It would be valuable to include such comparisons to provide a more comprehensive evaluation of your proposed model's performance.
Some typos are scattered in the paper. Please correct them.
Round 2
Reviewer 1 Report
This improvement article of yours is certainly better than the last one, and you have carried out more improvement work than we expected. However, there are some shortcomings as follows:
1. The formulas in the article are not neatly organized; the formula numbers should be aligned throughout.
2. There are paragraph formatting errors in lines 192, 313, 325 and 349 of the article.
3. Authors are advised to place the list of symbols in an appropriate place before the introduction and to remove duplications in the article from the list of symbols in order to avoid redundancy.
4. It is recommended that figure 12 in the conclusion be placed in the first citation section to ensure that readers can better understand the content of the figure when reading the article.
5. It is suggested that the conclusions should be organized in subsections to highlight the main body of work in this article.
Minor editing of English is required.
Reviewer 2 Report
The paper has been greatly refined. For the final submission, please provide more information about the dynamic adaptability of the scheme in terms of coding and modulation.
The English is mostly satisfactory.
